# Industrial Use of Phosphate Food Additives: A Mechanism Linking Ultra-Processed Food Intake to Cardiorenal Disease Risk?

**DOI:** 10.3390/nu15163510

**Published:** 2023-08-09

**Authors:** Mona S. Calvo, Elizabeth K. Dunford, Jaime Uribarri

**Affiliations:** 1Department of Medicine, Division of Nephrology, Icahn School of Medicine at Mount Sinai, New York, NY 10029, USA; mscalvo55@comcast.net; 2The George Institute for Global Health, University of New South Wales, Sydney, NSW 2042, Australia; edunford@georgeinstitute.org.au; 3Department of Nutrition, Gillings Global School of Public Health, The University of North Carolina at Chapel Hill, Chapel Hill, NC 27599, USA

**Keywords:** ultra-processed food, phosphorus intake, inorganic phosphate additives, NOVA food processing classification, cardiorenal disease, chronic kidney disease, cardiovascular disease, hormone dysregulation, FGF-23, PTH

## Abstract

The consumption of ultra-processed food (UPF) keeps rising, and at the same time, an increasing number of epidemiological studies are linking high rates of consumption of UPF with serious health outcomes, such as cardiovascular disease, in the general population. Many potential mechanisms, either in isolation or in combination, can explain the negative effects of UPF. In this review, we have addressed the potential role of inorganic phosphate additives, commonly added to a wide variety of foods, as factors contributing to the negative effects of UPF on cardiorenal disease. Inorganic phosphates are rapidly and efficiently absorbed, and elevated serum phosphate can lead to negative cardiorenal effects, either directly through tissue/vessel calcification or indirectly through the release of mineral-regulating hormones, parathyroid hormone, and fibroblast growth factor-23. An association between serum phosphate and cardiovascular and bone disease among patients with chronic kidney disease is well-accepted by nephrologists. Epidemiological studies have demonstrated an association between serum phosphate and dietary phosphate intake and mortality, even in the general American population. The magnitude of the role of inorganic phosphate additives in these associations remains to be determined, and the initial step should be to determine precise estimates of population exposure to inorganic phosphate additives in the food supply.

## 1. Introduction

There is a negative health perception of highly industrially processed foods and ultra-processed foods (UPFs) and beverages, which now dominate Western dietary patterns. This unhealthy assessment is reinforced by an ever-growing number of studies linking UPF consumption to increased disease risk [1], including obesity in adults and children [2,3], cardiovascular disease [4,5,6], chronic kidney disease [7], type 2 diabetes [8,9], cancer [10,11], neurologic disorders [12,13], and mortality [14,15]. Despite the flood of epidemiology studies exploring the association of high UPF consumption with the increased risk of non-communicable disease, little progress has been made toward establishing possible mechanism(s) through which high UPF consumption may promote chronic disease risk. Understanding how industrial food processing may generate possible biological mechanisms promoting these observed associations with non-communicable chronic disease is paramount to the safety and health of our modern food supply [16,17].

The NOVA system of classification of food based on the degree of processing subjectively characterizes a UPF by its use of industrial formulations, the presence of specific ingredients not commonly used in home food preparation, and reliance on the addition of processing ingredients (additives and substances) with specific technical functions, such as the industrial use of preservatives, sweeteners, color additives, flavors and spices, flavor enhancers, nutrients, emulsifiers, leavening agents, anticaking agents, stabilizers, and thickeners [18]. The NOVA system represents a transition from the traditional use of evaluating the healthy characteristics of dietary patterns previously based on nutrient content status to this vague, subjective method of determining the degree of processing and has met with considerable controversy [19]. Action to determine the variability by investigators using the NOVA classification and its impact on health risk has been taken, but more comparisons and refinement of the definition of UPF are warranted [20]. Given the current global application with some variations in the use of the NOVA classification system to evaluate dietary patterns and health risks, many investigators conclude that a standardization of the NOVA classification method is in order [21]. Identifying the presence of food additives listed in the *Ingredients* label of packaged foods and beverages has been proposed as a more objective method of identifying UPF and may prove to be an important method of identification of UPF with stronger associations to mechanisms by which industrial additives contribute to chronic disease risk [22]. 

## 2. Ultra-Processed Food, Cardiorenal Disease, and Phosphate Additives

### 2.1. Inorganic Phosphate Additives

There are over 3967 approved industrial food additives and substances in the FDA food inventory list (http://www.fda.gov/food/food-ingredients-packaging/overview-food-ingredients-additives-colors accessed on 7 July 2023); clearly focusing on just one individual additive or a class of food additives for research into the possible mechanism(s) through which additives in UPF may impair health is a daunting task. The objective of this review is to explain the main obstacles that must be overcome to understand and validate any mechanisms involved in the negative health effects of UPF consumption. Our decision to focus on one additive group sprang from our current understanding and past research of dietary factors impacting cardiorenal health and the key dietary factors that impact the risk and progression of cardiorenal risk. We have shown that excessive use of phosphate-containing food additives associated with the industrial processing of foods provides a plausible mechanism through which the high consumption of UPF can lead to an increased risk of renal and cardiovascular disease progression and mortality [23]. Any significant association between the high consumption of phosphate additives in UPF and disease risk also necessitates the reassessment of their safe use under current conditions of use, not a simple task given the obstacles faced in establishing the specificity and strength of the association with observed health risks. Safety assessment of any added substance to the US food supply involves several requirements that include demonstrating distribution in the food supply and population exposure estimates using dietary surveys to show actual or potential consumption levels. Exposure estimates are the fundamental data needed to link a UPF mechanism to disease risk or potential harm [24]; however, their food additive components present different problems in establishing estimates of exposure and possible association with a mechanism affecting disease risk. Approaches to exposure estimates needed to confirm safety emphasize the notion that substances tested in the past for their safe use under very different conditions may need to be re-evaluated for their safe use in the highly processed foods we eat today. With this objective, we examined a specific class of additives, inorganic phosphates (the mono-, di-, tri-, and poly-phosphate salts of phosphoric acid), numbering about 28 FDA-approved frequently used inorganic substances, largely composed of calcium, magnesium, sodium, and potassium salts, each with many technical functions (Appendix A).

### 2.2. Multiple Approved Technical Functions Complicate Study of Phosphate Additives and Regulation of Use

Undeniably, the frequent industrial use of more than 50 different organic and inorganic FDA-approved phosphate-containing additives to industrially processed foods significantly improves their palatability, taste, texture, nutritional value, safety, and other desirable qualities that are attributed to their FDA-approved technical functions. Figure 1 below shows the many FDA-approved technical functions and the number of phosphate additives with each approved function for use in processed foods. For most inorganic phosphate salt additives, the amount that can be added during processing is largely unregulated by FDA. 

These phosphate additives have GRAS (generally regarded as safe) status that limits the amount added to foods to be determined by a guide known as GMP (good manufacturing practice), which leaves the amount added to a food to the discretion of the manufacturer. The extent of use across all categories of foods is attributed to the multiple and varied FDA-approved technical functions required by all substances added to the US food supply [25]. Inorganic phosphate additives have multiple approved technical functions, allowing several to be used in a product or greater addition of one used for several different functions, all contributing to phosphate intake. The industrial use of various food additives is a hallmark of UPF, and 60% of foods purchased by Americans contain food additives (a 10% increase in use since 2001) [26]; thus, narrowing our focus to inorganic phosphate additives with known health risks is a logical starting point to examining possible mechanisms of action in UPF.

### 2.3. Exposure Estimates for Inorganic Phosphate Additives

A significant limitation to estimating individual or cumulative inorganic phosphate additive exposure in the US population is attributed to the lack of published information indicating which foods contain them and at what level and the inability to link this information to nutrition surveys capturing usual food consumption. Since it is not feasible to connect exposure data to survey intake data at this time in the US, reasonably accurate but unquantified exposure to phosphate additive intake from UPF can be estimated from records of grocery store purchases, as shown recently [26,27]. To estimate exposure to phosphate additives, we examined ingredient labels of US household packaged products from the top 25 food and beverage manufacturers to identify the total number of food products containing phosphate additives across USDA’s 23 food categories. USDA identifies 6 food categories contributing the majority of total P intake (81%) [28]; however, this is thought to be mostly natural P, as the additive contribution is rarely included. Using category-level sales data as a proxy for the actual intake of foods with phosphate additives, we determined the percentage of foods that contained phosphate additives in the 6 categories (Figure 2 below). 

More than 50% of bread, processed meats, and ready-made meals contained phosphate additives. USDA survey intakes of total P are thought to significantly underestimate total phosphate intake when phosphate additive use is not included, and evidence of this underestimation was shown in over 25 clinical studies [23]. The contributions to total phosphate intake from phosphate food additives are often referred to as “hidden phosphorus,” and this source matters when managing dietary intake in patients with chronic kidney disease (CKD) [29]. The USDA Branded Foods Products Database (BFPD) contains label information about food and beverage content for foods sold in the US. Picard et al. [30] reported that of the 3466 foods in the BFPD, 52% contained inorganic and organic phosphate additives, but only a small fraction listed phosphate content on the label. Relative to the inorganic phosphate added, the BFPD showed only 3% of the products displayed phosphate content, yet 20% contained inorganic phosphate additives. The US BFPD does not show all available foods containing additives, unlike recent reports from the French NutriNet-Santé survey reporting the percentage of participants consuming foods with additives, including the top 11 phosphate additives, and the percentage of consumers with intakes exceeding their acceptable daily intake (ADI) of 40 mg/kg bw/d [24]. Relative to the US, the French participants in this survey consumed more minimally processed foods; thus, the intake of ultra-processed foods with inorganic phosphate additives is likely much higher in the US. These findings of high phosphate-additive content, together with the reports of a significant trend across the last two decades of increasing UPF consumption for most food categories in the US, predict a likely increase in total phosphate intake from UPF currently in the marketplace [31]. The estimated consumption of UPF represents about 50 to 70% of energy intake across all races/ethnicities and socioeconomic groups in the US [32]. 

### 2.4. Is There a Potential Mechanism for Excess Inorganic Phosphate Additives to Promote Cardiorenal Disease Risk?

Inorganic phosphate additives contribute to an excessive intake of phosphate with increasing recent evidence of association with adverse health effects in the otherwise healthy general population [23]. Inorganic phosphate food additives represent an unknown dietary source of rapidly and efficiently absorbed phosphate in UPF [29], contributing to high total phosphate intakes that typically exceed dietary requirements and daily recommended intakes by 2–3 fold across adult age, sex, and race/ethnicity. A critical fact is that natural organic phosphate in foods linked to protein, lipids, and other natural components of food differ from phosphate in food additives by the speed and efficiency with which phosphate is absorbed and acutely raises serum concentration [23], while inorganic phosphorus in food additives is rapidly and more completely absorbed since it is not dependent on the speed of digestion. Inorganic phosphate additives or phosphate salts (but to a lesser degree, organic phosphate additives bound to choline and mono- and di-starches) release phosphate faster than natural organic phosphates in minimally processed meat and other foods. An acute oral inorganic phosphate additive load can represent a large phosphate burden, abruptly raising serum concentrations postprandially and acutely increasing the potential to disrupt hormonal mechanisms relied upon to maintain phosphate homeostasis [33]. 

The health hazard of a high dietary phosphate burden, particularly from inorganic phosphate additives, to patients with CKD and the association with high intakes to risk of bone loss, cardiovascular disease (CVD), and all-cause mortality is well known [23,33,34,35], but little is known about the effects of sustained excess phosphate intakes owed to the general population’s higher preference for UPF containing phosphate food additives. Few population studies examined the health effects of total dietary phosphate intake by healthy individuals, and all were limited by the underestimation of total phosphate intake without knowing the contribution of phosphate additives [36,37,38]. Despite this limitation, the 2014 investigation into the effects of phosphate intake in the general US population of healthy adults (n = 9686, aged 20 to 80 years) without diabetes, cancer, kidney, or cardiovascular disease, followed prospectively as part of the US National Health and Nutrition Examination Survey (NHANES, 1988–1994), showed that higher total dietary phosphate intakes were associated with an increased risk of all-cause mortality [37]. Alarmingly, the increased mortality risk started at a dietary intake of 1400 mg phosphate/d. This is a sobering realization, as typical intake estimates from recent NHANES (2015 to 2018) reported usual mean daily total intakes of 1596 mg for men >19 years (n = 4850) and 1195 mg for females (n = 4963), and half or more of the US adult healthy population consumed phosphate at 2 to 3 times more than their requirement of 580 mg. Whether repeating this analysis with recent NHANES data will show the same relationship is unclear but mean total phosphate intakes remain under-reported in the USDA nutrient content database linked to NHANES [23]. Consistent with these findings, Cai et al. [39] reported high UPF consumption associated with a decline in renal function among generally healthy adults in the Netherlands, but the role of phosphate additives in UPF-induced toxicity remains unclear.

### 2.5. Evidence of Hormonal Disruption from Dietary Interventions Studies with Foods Containing Phosphate Additives

Animal studies and clinical studies in CKD and dialysis patients have demonstrated that high dietary phosphate intake can induce negative effects by disrupting the tightly controlled endocrine regulation of phosphate. Table 1 summarizes key studies examining the consumption of a variety of market-available processed foods with the known addition of inorganic phosphate food additives [40,41,42,43,44,45,46,47,48,49,50,51,52,53]. 

These dietary studies reported outcomes associated with physiologic changes promoting cardiorenal disease. A few recent examples of these study outcomes, when various types of UPF were consumed, include the following: (1) the habitual consumption of highly processed instant ramen noodles significantly raised serum phosphate concentration [50]; (2) higher consumption of sausage and deli meats processed with phosphate additives were shown to be associated with higher carotid intimal thickness in middle-aged Finns [48]; (3) added phosphorus intake in foods relative to natural sources was consistently and inversely associated with HDL cholesterol in men and women [51]; and (4) added phosphate with higher bioavailability, but not natural phosphate, in catfish was negatively linked to estimated glomerular filtration rate (eGFR) in the Jackson Heart Study [52]. The earliest studies examining the hormonal effects of high phosphate-additive consumption from grocery market foods, now decades old, were conducted in healthy young adults and mimicked typical low-calcium diets consumed over several weeks [40,41,42,43]. High phosphate-additive consumption resulted in increased biomarkers of parathyroid hormone (PTH) action [40] and, with longer study duration, persistent elevation in PTH with a decrease in the active metabolite of vitamin D (calcitriol) that is usually activated by PTH to correct an acute decrease in serum ionized calcium concentration [41,42]. Select studies focusing on the intake of carbonated cola beverage, which requires the use of phosphoric acid in processing, are summarized in Table 2 and show secondary hyperparathyroidism and low calcium intake with the displacement of milk consumption with its high calcium content by high cola consumption [54,55,56,57,58,59,60]. 

These studies were among the first to show that high inorganic phosphate additives in food can disrupt the hormonal regulation of calcium and phosphorus. Kemi et al. [43] demonstrated a similar elevated PTH in a large cross-sectional study of postmenopausal Finnish women consuming high-phosphate, low-calcium diets rich in phosphate additives. This study demonstrated a potential mechanism, a typical low dietary calcium-to-phosphate ratio in highly processed foods, which disrupts the hormonal regulation of these minerals. 

### 2.6. Role of Excess Phosphate Intake in the Endocrine Disruption of Phosphate Homeostasis Leading to Cardiorenal Disease and Mortality

The studies presented in Table 1 and Table 2 represent important evidence that foods processed with inorganic phosphate additives impact the hormones regulating phosphate balance, but with the newer awareness of other phosphorus-regulating hormones, researchers have to use carefully controlled oral intakes of phosphate salts added to experimental diets, an approach known as phosphate loading. Recently, Volk and colleagues [61] studied the effect of inorganic phosphate additives on cardiorenal risk factors and the well-established phosphate regulating hormone, fibroblast growth factor-23 (FGF-23) secreted by skeletal osteocytes in response to high serum phosphate. The 8-hour post-prandial hormone response in healthy young men to acute inorganic phosphate oral loading showed a significant increase in PTH, but not FGF-23, in this short time frame.

It is well-established in CKD and healthy subjects that habitual high-phosphate diets or acute oral phosphate salt loads of longer study duration can induce the release of two endocrine hormones, PTH from the parathyroid glands and FGF-23 from bone [62,63,64,65]. Sustained elevation in either one of these hormones can exert significant pathogenic cardiovascular, renal, and bone effects directly or synergistically with those induced by high phosphate concentration in tissues (Figure 3 below) [65]. 

These findings are consistent with earlier work showing FGF-23 is not quick to respond to acute variations in serum phosphate [61]. In contrast, studies in healthy adults exposed to very high inorganic phosphate intake [62] or observed responses to high-phosphate diets over several days showed modest increases in FGF-23 and decreases in the active metabolite of vitamin D [63]. The duration of exposure to high-phosphate additives and the level of inorganic phosphate seem to influence the nature of the hormonal response and likely the severity of the hormonal disruption to phosphate balance with sustained exposure [65]. A recent study from Denmark showed a significant association between urine phosphate and CVD, further suggesting the potential role of phosphate additives (significant contributors to total urine phosphate) in causing disease [66]. Increasing phosphate intake leads, at least transiently, to increased serum phosphate, which in turn, either directly through tissue calcification or indirectly through the release of PTH and FGF23, can affect bone, the cardiovascular system, and kidneys. FGF23 has been shown to have a direct effect on myocardial function, and it has been associated with valvular and vascular calcifications in dialysis patients, as illustrated in Figure 3. The combination of renal, cardiovascular, and bone effects eventually contributes to increased morbidity and mortality in advanced CKD and dialysis patients [66,67,68,69]. As documented above, enough data suggest a similar, although more subtle, effect of very high phosphate intake on phosphate metabolism and chronic disease mechanisms in the general population [70].

## 3. Regulatory Action Needed to Ensure Safety of Industrial Inorganic Phosphate Food Additives in UPF

Finding a significant association between high dietary phosphate intake and all-cause mortality and disruption of the endocrine regulation of phosphate in healthy people, not just in CKD and dialysis patients, is compelling evidence of the need to develop new public health and regulatory policies to contain this escalating but generally ignored dietary source of excessive phosphate intake. Globally, regulatory agencies have an important role to play. As UPF intake continues to grow, so has the evidence that a high percentage of these foods contains phosphate additives contributing to excessive phosphate intake [23]. When petitions from concerned physicians and health providers brought this issue to the attention of the European Food Safety Authority (EFSA), it responded in part by reassessing the safety of inorganic phosphate additive contribution to the total phosphate intake. After careful review, EFSA officially decreased the level of the phosphate additive intake considered safe, the acceptable daily intake (ADI), from 70 to 40 mg/kg bw/d and concluded that inorganic phosphate additives contribute 30 to 60% of total phosphate intake [71]. This policy change is critical to the safety assessment used by regulatory agencies to evaluate additive and ingredient safety in the general European population [24]. 

The lower ADI is not an effective regulatory guide in the US since the average body weight of most Americans is greater than that of Europeans and the ADI is specific to a food additive group of those derived from phosphoric acid, which have a specific mechanism of action that impacts health. The problem with phosphate additive use concerns excessive total phosphate intake in the US, which can disrupt mineral regulation. In the US, the regulatory reference against which the FDA evaluates the safety of excess intake of a nutrient is the upper tolerable level of dietary intake or UL. By definition, the UL is the level of nutrient intake beyond which adverse health effects may occur [72]. Combined with accurate estimates of exposure in the general population and dietary guidelines for nutrient requirements (EAR and RDA), the UL is used as a biomarker to evaluate the adequacy or excess of nutrient intake. An example of this is shown in Figure 4 below, presenting recent NHANES survey data across percentile intakes of total phosphate for adult men and women relative to their recommended daily intake (RDA) and estimated average requirement (EAR). Even with a UL of 4000 mg phosphate/d, which is almost seven times greater than the average adult requirement, there is a need for more compelling evidence of harm with current phosphate intakes that can warrant the reassessment of the phosphorus UL established in 1997 by the National Academy of Science Engineering and Medicine [72,73].

In sharp contrast to our evidence that phosphate additive use and consumption are increasing and the belief that it is a key contributor to excess intake, Fulgoni et al. [74] used a complex approach reliant on several assumptions to estimate added phosphate exposure to conclude that “Consumption of added phosphorus has decreased over the past few decades, possibly due to increased demand for foods with less additives/ingredients but may also be due to inaccurate phosphorus values in nutrition databases”. Grocery sale label information, while not without limitation, suggests the latter is most likely the case [26]. This highlights the need for critical regulatory action to update the nutrient content databases to reflect phosphate additive exposure and contribution to total intake because of changes in processing use or declared phosphate content on the food label. The regulatory changes needed for accurate phosphate content of foods provided in the databases require the FDA to mandate food manufacturers to list the phosphate content of foods on the nutrition facts label as the agency has done for potassium and vitamin D [23]. 

## 4. Future Considerations for At-Risk Populations to Minimize Phosphate Intake

Specific to inorganic phosphate food additives’ role in adverse health events with high UPF intakes, it remains unclear if these additives work in unison with other attributes of UPF, such as high added sugar, salt, or fat; low fiber; or even overconsumption of energy. From this perspective, the most effective strategy to limit disease risk may be to simply replace UPFs containing phosphate additives with minimally processed foods that do not contain these additives [75]. This was the effective approach taken in two studies focused on slowing the progression of CKD, in which patients were taught to avoid these foods [44] or replace them with similar products without phosphate additives [49]. An alternative strategy is to focus on specific food categories, such as processed meats, whose manufacturers are developing products with alternatives to phosphate additives [76]. Examples of this type of product reformulation are currently on the market, such as a traditionally processed graham cracker leavened with calcium phosphate compared to a reformulated graham cracker that uses tartaric acid as a leavening agent, thereby developing a product better suited to CKD patients.

The obstacles encountered in efforts to identify specific food additives in UPF that may play a role in the multiple observed risks for non-communicable diseases likely differ among classes and technical functions of food additives and their specific physiologic effects. The outcome measures to use that appropriately assess harm and adverse health effects implicating a disease mechanism vary widely for different additives. Individually tailored diets that limit those UPFs known to contain major dietary risk factors, as shown in the examples for CKD patients, fits the concept of “food as medicine” and can be used to potentially prevent and treat CKD patients [77]. Capozzi et al. wisely suggest that before reformulating the UPF in the global food supply associated with adverse health, we must first explore needed changes from the perspective of the disease in question, initially examining known dietary factors that can impact the disease state [78]. Further research is warranted to explore food additive distribution across UPF food categories, trends toward increasing use, availability/feasibility of alternatives, and the socioeconomic impact of the transition to minimally processed foods. Research is needed to determine the effects of the early introduction of food additives in baby foods, the food category showing the highest increase in additive use since 2001 [26,27]. Caution is also warranted at this early stage of discovery for disease risks with UPF consumption with implications of causation without supporting evidence; the foremost concern should be with individuals at greatest disease risk, as we have explained for CKD and CVD patients, and not focused on the reformulation of the vast categories of UPF in the food supply [78]. 

## Figures and Tables

**Figure 1 nutrients-15-03510-f001:**
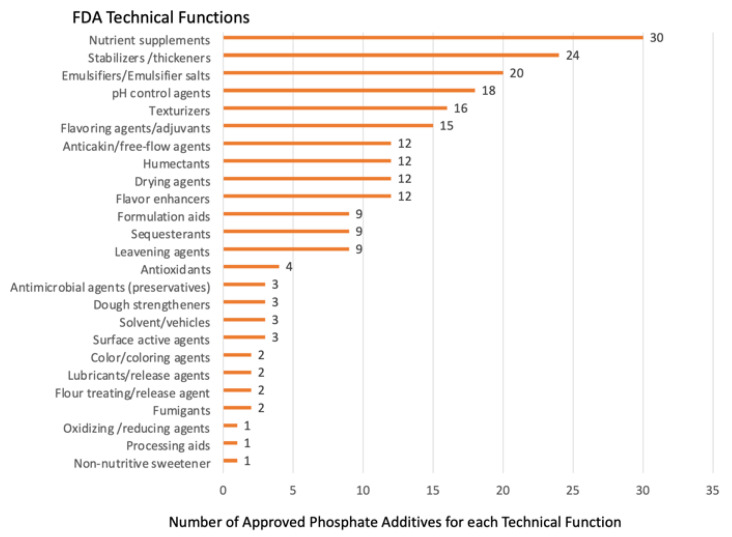
The number of phosphate additives approved by FDA for use in processing (X-axis) to function as one of the 26 different technical functions listed on the Y-axis.

**Figure 2 nutrients-15-03510-f002:**
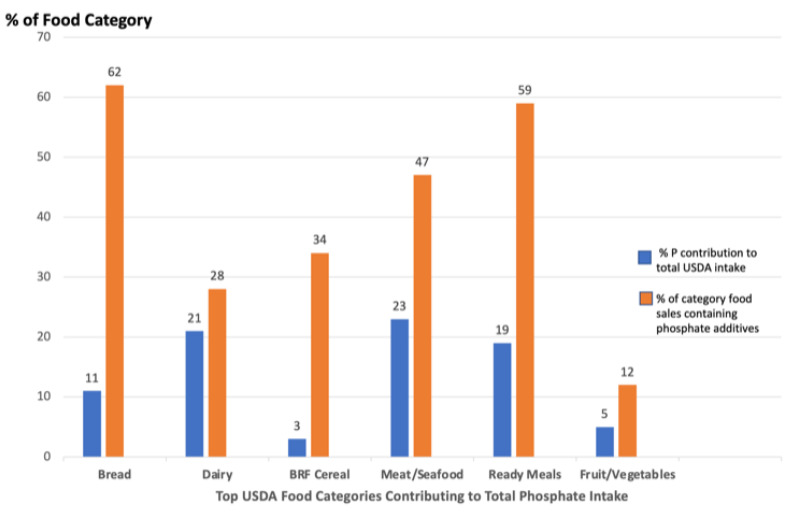
The conceptualization of the extent of use of phosphate additives in grocery food categories from the top 25 US grocery manufacturers and the inconsistency with USDA estimates of food categories contributing to total phosphate intake. Bread and ready-made meals (e.g., frozen dinners and pizza) contain the most phosphate food additives but seemingly are minor contributors to USDA estimates of total phosphate intake.

**Figure 3 nutrients-15-03510-f003:**
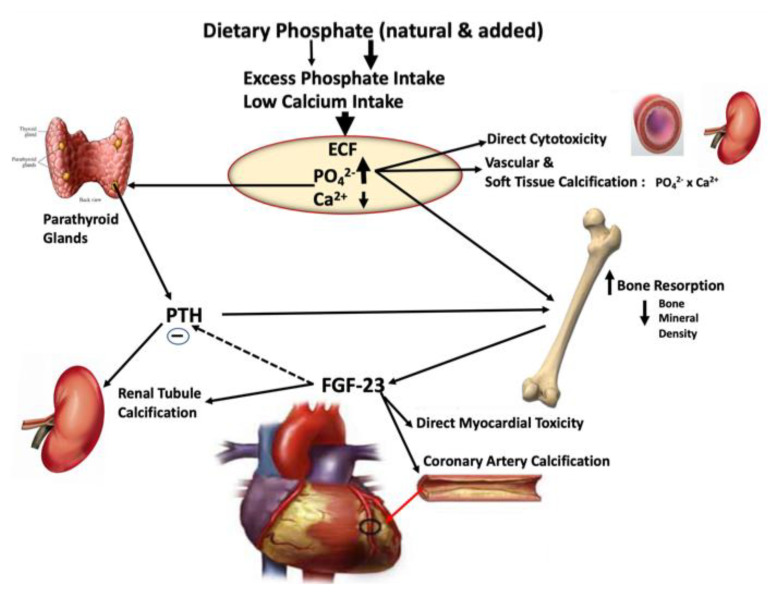
The known dietary path of excess phosphate intake leading to hormonal dysregulation of phosphorus and calcium homeostasis and the ultimate progression to CKD, CVD, and mortality. Greater and more rapid absorption of added phosphates coupled with the typical low calcium intake in the US diet produces a mineral imbalance in extracellular fluid (ECF) with phosphate in excess and a decrease in ionized calcium. This imbalance stimulates PTH secretion from the parathyroid glands, which directly stimulates bone resorption and triggers bone osteocytes to release FGF-23. Hyperphosphatemia can directly stimulate FGF-23 release, cytotoxicity, and soft-tissue calcification; however, it is FGF-23 that is thought to have the most damaging action when concentrations are sustained over time. FGF-23 directly induces myocardial toxicity, promotes coronary artery and renal tubule calcification, inhibits PTH action, thereby inhibiting calcitriol synthesis needed to correct low serum calcium, and continues to stimulate bone resorption exacerbating soft-tissue calcification.

**Figure 4 nutrients-15-03510-f004:**
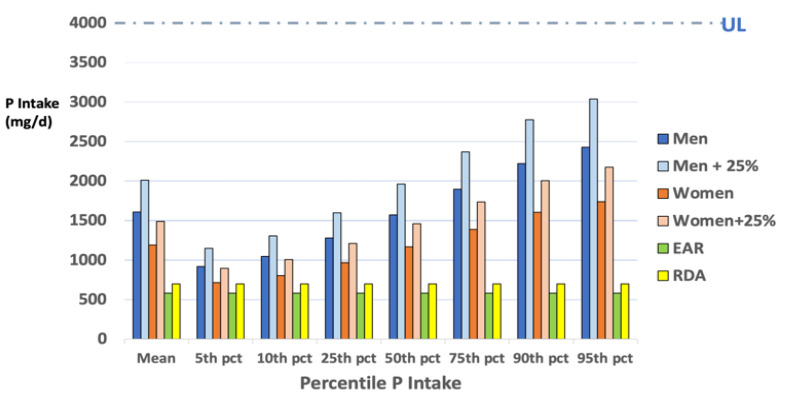
The usual phosphorus intake from food and beverages for adults ≥19 y plotted across percentiles of intake estimated from NHANES 2013–2016 survey data and shown by the solid blue (men) and orange (women) bars. Lighter blue and orange bars represent calculated intakes of phosphorus increased by 25%, which conservatively estimates total intakes, including a postulated unaccounted phosphate additive use in the USDA nutrient database linked to NHANES surveys. The green and yellow bars represent the adult EAR and RDA for phosphorus (580 and 700 mg/d, respectively), while the dashed line at the top represents the UL or upper level of safe intake for phosphorus for all adult ages, gender, and body weights at 4000 mg/d. Data source: USDA Agricultural Research Service, 2019. Usual Nutrient Intake from Food and Beverages by Gender and Age, from What We Eat in America, NHANES 2013–2016. Available: http://www.ars.usda.gov/nea/bhnrc/fsre accessed on 10 April 2023.

**Table 1 nutrients-15-03510-t001:** Adverse Outcomes from Consuming Foods High in Phosphate Additives.

Studies Using Market-Available Foods Containing Phosphate Additives [Reference]	Study Outcomes
Bell RR et al., 1977 [40]	Elevated parathyroid hormone biomarkers with high-phosphate additives in foods, >200 mg P/day, dietary Ca:P ratio = 0.35
Calvo MS et al., 1988 [41]	Elevated parathyroid hormone after 2 weeks of grocery store foods with high-phosphate additives and low calcium content, dietary Ca:P ratio = 0.3
Calvo MS et al., 1990 [42]	Parallel study design with test diet high in phosphate additives, low calcium content from purchased foods showing persistent elevation in parathyroid hormone but decrease in calcitriol after consuming for 4 weeks, Ca:P ratio = 0.3
Kemi VE et al., 2009 [43]	A cross-sectional population study in healthy Finnish women showed higher parathyroid hormone concentrations in those consuming diets with Ca:P ratio = 0.56.
Sullivan C et al., 2009 [44]	Avoidance of foods with phosphate additives reduced serum phosphorus in dialysis patients.
Itkonen ST et al., 2013 [45]	Greater intakes of foods high in phosphate additives were associated with greater carotid intima thickness.
Gutiérrez OM et al., 2015 [46]	After a week of low phosphate additive diets (Ca:P = 0.77), a week on a diet high in phosphate additive (Ca:P = 0.47) increased fibroblast growth factor-23 and markers of bone metabolism.
Moore, LW, et al., 2015 [47]	Dairy and cereal grain products with phosphate food additives significantly increased serum phosphorus in early chronic kidney disease.
Itkonen, ST, et al., 2017 [48]	Phosphate intake in women, not in men, was negatively associated with bone formation markers.
De Fornasari ML et al., 2017 [49]	Phosphate additive–free food consumption reduced hyperphosphatemia and parathyroid hormone concentrations in dialysis patients.
Saito Y et al., 2021 [50]	Consumers of instant ramen noodles (high in phosphate additives) had higher serum phosphate.
Fulgoni K et al., 2022 [51]	Added phosphate intake was consistently inversely associated with HDL cholesterol in both men and women (NHANES data).
Duong CN et al., 2022 [52]	Phosphorus contents in foods were designated as natural or added, and intake was weighted by their bioavailability (used an algorithm based on literature). Added, but not natural, phosphorus was negatively linked to the estimated glomerular filtration rate in the Jackson Heart Study.
Moroșan E et al., 2023 [53]	Dialysis patients treated with personalized nutrition therapy substituting processed high phosphate-additive foods with low phosphate-additive foods and use of phosphate binders significantly reduced phosphatemia after 60 days of treatment.

**Table 2 nutrients-15-03510-t002:** Adverse Health Outcomes from Consuming Cola Beverages *.

Studies Examining Cola Intake [Reference]	Study Outcome
Mazariegos-Ramos E et al., 1995 [54]	Hypocalcemia in children was directly associated with the consumption of soft drinks containing phosphoric acid.
Fernando GR et al., 1999 [55]	Hypocalcemia in postmenopausal women was associated with the consumption of soft drinks processed with phosphoric acid.
Kristensen M et al., 2005 [56]	Increased serum phosphate, parathyroid hormone, bone turnover marker osteocalcin, and serum and urinary markers of bone resorption were among the acute effects of replacing milk with cola beverages for 10 days.
Tucker KL et al., 2006 [57]	Cola beverage (containing phosphoric acid) consumption was associated with low bone mineral density in older women.
Guarnotta V et al., 2019 [58]	Daily cola consumption was associated with hypocalcemia.
Gallagher JC 2019 [59]	Carbonated beverage and cola consumption in young Korean males was inversely associated with the whole body, whole femur, and femoral neck bone mineral density.
Kim YA and Yoo JH 2020 [60]	A cross-sectional population study of cola consumption in young Korean males reported inversely associated cola intake and whole body, whole femur, and femoral neck bone mineral density.

* Cola beverages are processed with phosphoric acid, which can serve a number of critical technical functions, such as a flavor enhancer, acidulant, or color enhancer.

## Data Availability

Not applicable.

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
