# Peer review of "Industrial Use of Phosphate Food Additives: A Mechanism Linking Ultra-Processed Food Intake to Cardiorenal Disease Risk?"

_nutrients, 2023, doi:10.3390/nu15163510_

Round 1
Reviewer 1 Report
Comments to the authors:
-Abstract should be shortened. Furthermore, in the abs authors should say something regarding the mechanisms through which phosphates would increase cardio-renal disease risk
-Table 1 should be moved to supplementary materials
-In Figure 1 it's not clear how many additives are approved for each function. Please reorganize the figure. Titles of the axis are missing.
Reviewer 2 Report
This review discusses the adverse health effect of phosphate additives in ultra processed food with some focus on current regulatory practices.
1. The main correction needed in this manuscript is that figures and tables need to have legends below them (as per standard scientific publication practices). Certain figures also need appropriate labelling. For eg: Figure 1 does not have X and Y axis labelled. Number of items on the Y axis is different than number of bars in the chart. Large and bold fonts on top of figures make them look like a poster presentation. It is of utmost importance that each table and figure have their individual description below them.
2. Grammatical and typographical errors are present throughout the manuscript that needs correction. Some include : Line 17 - extra space before 'Inorganic', Line 46 - extra space before 'the presence', line 48 - extra space before 'use of', Line 70 - p in 'perspective' is capitalized.
3. Some abbreviations requires their full form when they are used for the first time in the manuscript or even once. For eg: eGFR, CTX, NTX etc.
English is fine. Grammatical and typographical corrections needed throughout.
Round 2
Reviewer 2 Report
Desired changes have been made. Manuscript can be accepted.